# One ticket to win them all: generalizing lottery ticket initializations across datasets and optimizers

**Ari S. Morcos**[*]
Facebook AI Research
arimorcos@fb.com

**Haonan Yu**
Facebook AI Research
haonanu@gmail.com

**Michela Paganini**
Facebook AI Research
michela@fb.com

**Yuandong Tian**
Facebook AI Research
yuandong@fb.com

## Abstract

The success of lottery ticket initializations [7] suggests that small, sparsified networks can be trained so long as the network is initialized appropriately. Unfortunately, finding these "winning ticket" initializations is computationally expensive. One potential solution is to reuse the same winning tickets across a variety of datasets and optimizers. However, the generality of winning ticket initializations remains unclear. Here, we attempt to answer this question by generating winning tickets for one training configuration (optimizer and dataset) and evaluating their performance on another configuration. Perhaps surprisingly, we found that, within the natural images domain, winning ticket initializations generalized across a variety of datasets, including Fashion MNIST, SVHN, CIFAR-10/100, ImageNet, and Places365, often achieving performance close to that of winning tickets generated on the same dataset. Moreover, winning tickets generated using larger datasets consistently transferred better than those generated using smaller datasets. We also found that winning ticket initializations generalize across optimizers with high performance. These results suggest that winning ticket initializations generated by sufficiently large datasets contain inductive biases generic to neural networks more broadly which improve training across many settings and provide hope for the development of better initialization methods.

## 1   Introduction

The recently proposed lottery ticket hypothesis [7, 8] argues that the initialization of over-parameterized neural networks contains much smaller sub-network initializations, which, when trained in isolation, reach similar performance as the full network. The presence of these "lucky" sub-network initializations has several intriguing implications. First, it suggests that the most commonly used initialization schemes, which have primarily been discovered heuristically [10, 14], are sub-optimal and have significant room to improve. This is consistent with work which suggests that theoretically-grounded initialization schemes can enable the training of extremely deep networks with hundreds of layers [13, 27, 29, 33, 35]. Second, it suggests that over-parameterization is not necessary during the course of training as has been argued previously [1, 2, 5, 6, 24, 25], but rather over-parameterization is merely necessary to find a "good" initialization of an appropriately parameterized network. If this hypothesis is true, it indicates that by training and performing inference in networks which are 1-2 orders of magnitude larger than necessary, we are wasting large amounts of

---

[*]To whom correspondence should be addressed

computation. Taken together, these results hint toward the development of a better, more principled initialization scheme.

However, the process to find winning tickets requires repeated cycles of alternating training (potentially large) models from scratch to convergence and pruning, making winning tickets computationally expensive to find. Moreover, it remains unclear whether the properties of winning tickets which lead to high performance are specific to the precise combination of architecture, optimizer, and dataset, or whether winning tickets contain inductive biases which improve training more generically. This is a critical distinction for evaluating the future utility of winning ticket initializations: if winning tickets are overfit to the dataset and optimizer with which they were generated, a new winning ticket would need to be generated for each novel dataset which would require training of the full model and iterative pruning, significantly blunting the impact of these winning tickets. In contrast, if winning tickets feature more generic inductive biases such that the same winning ticket generalizes across training conditions and datasets, it unlocks the possibility of generating a small number of such winning tickets and reusing them across datasets. Further, it hints at the possibility of parameterizing the distribution of such tickets, allowing us to sample generic, dataset-independent winning tickets.

Here, we investigate this question by asking whether winning tickets found in the context of one dataset improve training of sparsified models on other datasets as well. We demonstrate that individual winning tickets which improve training across many natural image datasets can be found, and that, for many datasets, these transferred winning tickets performed almost as well as (and in some cases, better than) dataset-specific initializations. We also show that winning tickets generated by larger datasets (both with more training samples and more classes) generalized across datasets substantially better than those generated by small datasets. Finally, we find that winning tickets can also generalize across optimizers, confirming that dataset- and optimizer-independent winning ticket initializations can be generated.

## 2 Related work

Our work is most directly inspired by the lottery ticket hypothesis, which argues that the probability of sampling a lucky, trainable sub-network initialization grows with network size due to the combinatorial explosion of available sub-network initializations. The lottery ticket hypothesis was first postulated and examined in smaller models and datasets in [7] and analyzed in large models and datasets, leading to the development of late resetting, in [8]. The lottery ticket hypothesis has recently been challenged by [20], which argued that, if randomly initialized sub-networks with structured pruning are scaled appropriately and trained for long enough, they can match performance from winning tickets. [9] also evaluated the lottery ticket hypothesis in the context of large-scale models and were unable to find successful winning tickets, although, critically, they did not use iterative pruning and late resetting, both of which have been found to be necessary to induce winning tickets in large-scale models. However, all of these studies have only investigated situations in which winning tickets are evaluated in an identical setting to that in which they generated, and therefore do not measure the generality of winning tickets.

This work is also strongly inspired by the model pruning literature as well, as the choice of pruning methodology can have large impacts on the structure of the resultant winning tickets. In particular, we use magnitude pruning, in which the lowest magnitude weights are pruned first, which was first proposed by [12]. A number of variants have been proposed as well, including structured variants [19] and those which enable pruned weights to recover during training [11, 37]. Many other pruning methods have been proposed, including greedy methods [22], methods based on variational dropout [21], and those based on the similarity between the activations of feature maps [3, 28].

Transfer learning has been studied extensively with many studies aiming to transfer learned representations from one dataset to another [17, 31, 34, 38]. Pruning has also been analyzed in the context of transfer learning, primarily by fine-tuning pruned networks on novel datasets and tasks [22, 37]. These results have demonstrated that training models on one dataset, pruning, and then fine-tuning on another datasets can often result in high performance on the transfer dataset. However, in contrast to the present work, these studies investigate the *transfer of learned representations*, whereas we analyze the *transfer of initializations* across datasets.

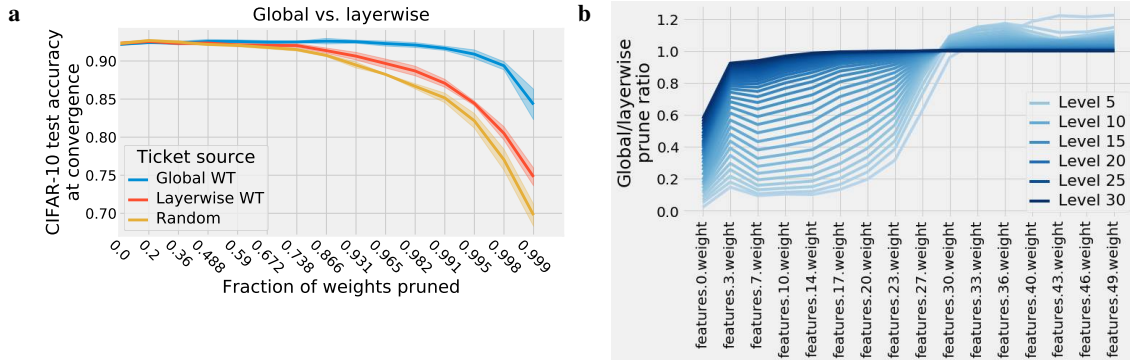

Figure 1: **Global vs. layerwise pruning.** (**a**) CIFAR-10 winning ticket performance for global pruning (blue) vs. layerwise pruning (red) along with a random ticket (red). Error bars represent mean ± standard deviation across six random seeds. (**b**) Ratio of global pruning rate to layerwise pruning rate at each convolutional layer in VGG19. Level represents the pruning iteration such that lighter blues represent lower overall pruning rates, while the darkest blue represents a 0.999 overall pruning rate.

## 3 Approach

### 3.1 The lottery ticket hypothesis

The lottery ticket hypothesis proposes that "lucky" sub-network initializations are present within the initializations of over-parameterized networks which, when trained in isolation, can reach the same or, in some cases, better test accuracy than the full model, even when over 99% of the parameters have been removed [7]. This effect is also present for large-scale models trained on ImageNet, suggesting that this phenomenon is not specific to small models [7]. In the simplest method to find and evaluate winning tickets, models are trained to convergence, pruned, and then the set of remaining weights are reset to their value at the start of training. This smaller model is then trained to convergence again ("winning ticket") and compared to a model with the same number of parameters but randomly drawn initial parameter values ("random ticket"). A good winning ticket is one which significantly outperform random tickets. However, while this straightforward approach finds good winning tickets for simple models and datasets (e.g., MLPs trained on MNIST), this method fails for more complicated architectures and datasets, which require several "tricks" to generate good winning tickets:

**Iterative pruning**  When models are pruned, they are typically pruned according to some criterion (see related work for more details). While these pruning criteria can often be effective, they are only rough estimates of weight importance, and are often noisy. As such, pruning a large fraction of weights in one step ("one-shot pruning") will often prune weights which were actually important. One strategy to combat this issue is to instead perform many iterations of alternately training and pruning, with each iteration pruning only a small fraction of weights [12]. By only pruning a small fraction of weights on each iteration, iterative pruning helps to de-noise the pruning process, and produces substantially better pruned models and winning tickets [7, 8]. For this work, we used magnitude pruning with an iterative pruning rate of 20% of remaining parameters.

**Late resetting**  In the initial investigation of lottery tickets, winning ticket weights were reset to their values at the beginning of training (training iteration 0), and learning rate warmup was found to be necessary for winning tickets on large models [7]. However, in follow-up work, Frankle et al. [8] found that simply resetting the winning ticket weights to their values at training iteration $k$, with $k$ much smaller than the number of total training iterations consistently produces better winning tickets and removes the need for learning rate warmup. This approach has been termed "late resetting." In this work, we independently confirmed the importance of late resetting and used late resetting for all experiments (See Appendix A.1 for the precise late resetting values used for each experiment).

**Global pruning**  Pruning can be performed in two different ways: locally and globally. In local pruning, weights are pruned within each layer separately, such that every layer will have the same fraction of pruned parameters. In global pruning, all layers are pooled together prior to pruning,

allowing the pruning fraction to vary across layers. Consistently, we observed that global pruning leads to higher performance than local pruning (Figure 1a). As such, global pruning is used throughout this work. Interestingly, we observed that global magnitude pruning preferentially prunes weights in deeper layers, leaving the first layer in particular relatively unpruned (Figure 1b; light blues represent early pruning iterations with low overall pruning fractions while dark blues represent late pruning iterations with high pruning fractions). An intuitive explanation for this result is that because deeper layers have many more parameters than early layers, pruning at a constant rate harms early layers more since they have fewer absolute parameters remaining. For example, the first layer of VGG19 contains only 1792 parameters, so pruning this layer at a rate of 99% would result in only 18 parameters remaining, significantly harming the expressivity of the network.

**Random masks**    In the sparse pruning setting, winning ticket initializations contain two sources of information: the values of the remaining weights and the structure of the pruning mask. In previous work [7–9], the structure of the pruning mask was maintained for the random ticket initializations, with only the values of the weights themselves randomized. However, the structure of this mask contains a substantial amount of information and requires fore-knowledge of the winning ticket initialization (see Figure A1 for detailed comparisons of different random masks). For this work, we therefore consider random tickets to have both randomly drawn weight values (from the initialization distribution) *and* randomly permuted masks. For a more detailed discussion of the impact of different mask structures, see Appendix A.2.

### 3.2   Models

Experiments were performed using two models: a modified form of VGG19 [30] and a ResNet50 [15]. For VGG19, structure is as in [30], except all fully-connected layers were removed such that the last layer is simply a linear layer from a global average pool of the last convolutional layer to the number of output classes (as in [7, 8]). For details of model architecture and hyperparameters used in training, see Appendix A.1.

### 3.3   Transferring winning tickets across datasets and optimizers

In order to evaluate the generality of winning tickets, we generate winning tickets in one training configuration ("source") and evaluate performance in a different configuration ("target"). For many transfers, this requires changing the output layer size, since datasets have different numbers of output classes. Since the winning ticket initialization is only defined for the source architecture and therefore cannot be transferred to different topologies, we simply excluded this layer from the winning ticket and randomly reinitialized it. Since the last convolutional layer of our models was globally average pooled prior to the final linear layer, changes in input dimension did not require modification to the model.

For standard lottery ticket experiments in which the source and target dataset are identical, each iteration of training represents the winning ticket performance for the model at the current pruning fraction. However, because the source and target dataset/optimizer are different for our experiments and because we primarily care about performance on the target dataset for this study, we must re-evaluate each winning ticket's performance on the target dataset, adding an additional training run for each pruning iteration. We therefore run two additional training runs at each pruning iteration: one for the winning ticket and one for the random ticket on target configuration.

## 4   Results

For all experiments, we plot test accuracy at convergence as a function of the fraction of pruned weights. For each curve, 6 replicates with different random seeds were run, with shaded error regions representing $\pm$ 1 standard deviation. For comparisons on a given target dataset or optimizer, models were trained for the same number of epochs (see Appendix A.1 for details).

### 4.1   Transfer within the same data distribution

As a first test of whether winning tickets generalize, we investigate the simplest form of transfer: generalization across samples drawn from the same data distribution. To measure this, we divided

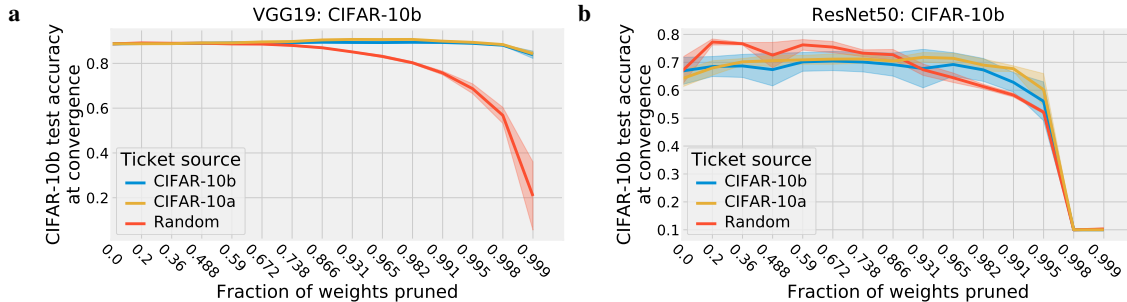

**Figure 2: Transferring winning tickets within the same data distribution.** CIFAR-10 was divided into two halves ("10a" and "10b"), each of which contained 25,000 total examples with 2,500 images per class. Winning tickets generated using CIFAR-10a generalized well to CIFAR-10b for both VGG19 (**a**) and ResNet50 (**b**). Error bars represent mean ± standard deviation across six random seeds.

the CIFAR-10 dataset into two halves: CIFAR-10a and CIFAR-10b. Each half contained 25,000 training images with 2,500 images per class. We then asked whether winning tickets generated using CIFAR-10a would produce increased performance on CIFAR-10b. To evaluate the impact of transferring a winning ticket, we compared the CIFAR-10a ticket to both a random ticket and to a winning ticket generated on CIFAR-10b itself (Figure 2). Interestingly, for ResNet50 models, while both CIFAR-10a and CIFAR-10b winning tickets outperformed random tickets at extreme pruning fractions, both under-performed random tickets at low pruning fractions, suggesting that ResNet winning tickets may be particularly sensitive to smaller datasets at low pruning fractions.

## 4.2 Transfer across datasets

Our experiments on transferring winning tickets across training data drawn from the same distribution suggest that winning tickets are not overfit to the particular data samples presented during training, but winning tickets may still be overfit to the data distribution itself. To answer this question, we performed a large set of experiments to assess whether winning tickets generated on one dataset generalize to different datasets within the same domain (natural images).

We used six different natural image datasets of various complexity to test this: Fashion-MNIST [32], SVHN [23], CIFAR-10 [18], CIFAR-100 [18], ImageNet [4], and Places365 [36]. These datasets vary across a number of axes, including grayscale vs. color, input size, number of output classes, and training set size. Since each pruning curve comprises the result of training six models (to capture variability due to random seed) from scratch at each pruning fraction, these experiments required extensive computation, especially for models trained on large datasets such as ImageNet and Places365. We therefore only evaluated tickets from larger datasets on these datasets to best prioritize computation. For all comparisons, we performed experiments on both VGG19 (Figure 3) and ResNet50 (Figure 4).

Across all comparisons, several key trends emerged. First and foremost, individual winning tickets which generalize across all datasets with performance close to that of winning tickets generated on the target dataset *can be found* (e.g., winning tickets sourced from ImageNet and Places365 datsets). This result suggests that a substantial fraction of the inductive bias provided by winning tickets is *dataset-independent* (at least within the same domain and for large source datasets), and provides hope that individual tickets or distributions of such winning tickets may be generated once and used across different tasks and environments.

Second, we consistently observed that winning tickets generated on larger, more complex datasets generalized *substantially better* than those generated on small datasets. This is particularly noticeable for winning tickets generated on the ImageNet and Places365 source datasets, which demonstrated competitive performance across all datasets. Interestingly, this effect was not merely a result of training set size, but also appeared to be impacted by the number of classes. This is most clearly exemplified by the differing performance of winning tickets generated on CIFAR-10 and CIFAR-100, both of which feature 50,000 total training examples, but differ in the number of classes. Consistently, CIFAR-100 tickets generalized better than CIFAR-10 tickets, even to simple datasets such as Fashion-MNIST and SVHN, suggesting that simply increasing the number of classes while keeping the dataset

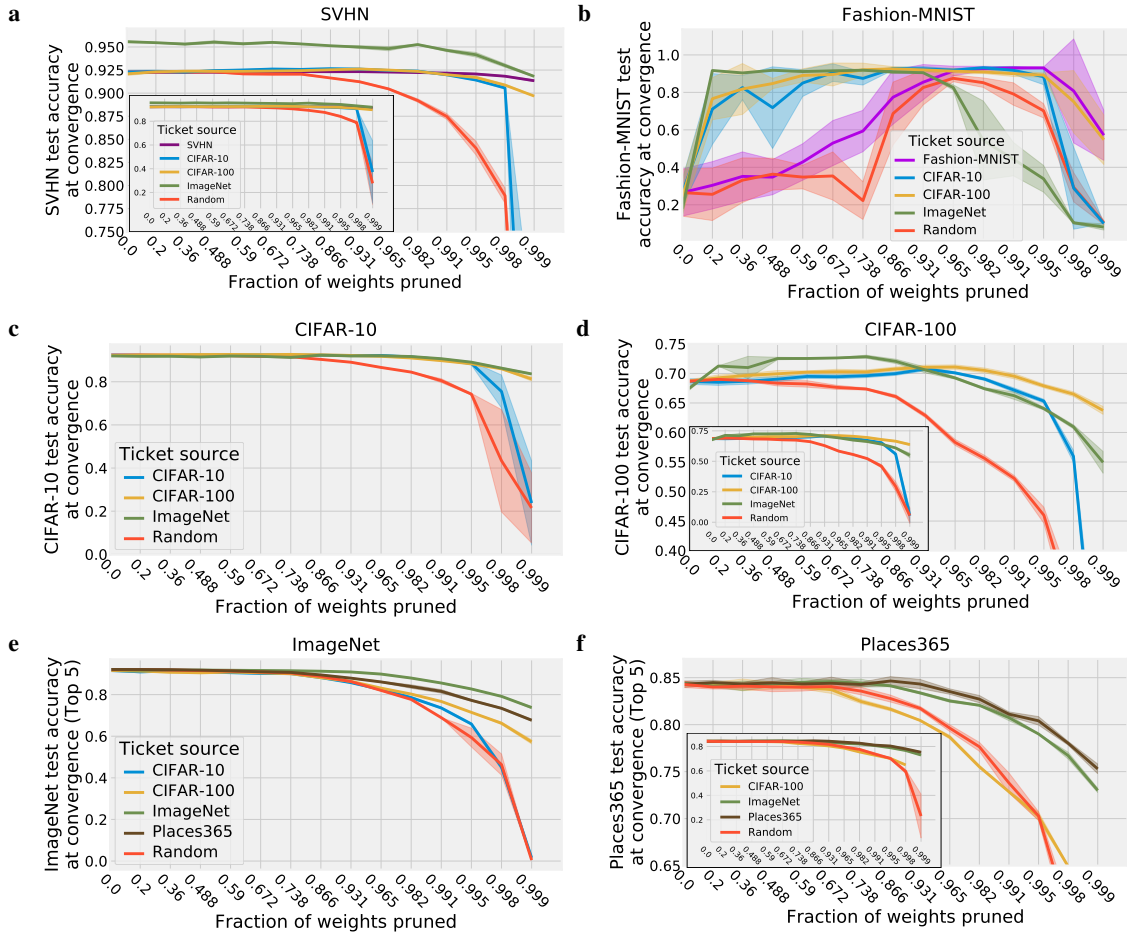

**Figure 3: Transferring VGG19 winning tickets across datasets.** Winning ticket performance on target datasets: SVHN (**a**), Fashion-MNIST (**b**), CIFAR-10 (**c**), CIFAR-100 (**d**), ImageNet (**e**), and Places365 (**f**). Within each plot, each line represents a different source dataset for the winning ticket. In cases where the y-axis has been narrowed to make small differences visible, full y-axes are provided as insets. Error bars represent mean ± standard deviation across six random seeds.

size fixed may lead to substantial gains in winning ticket generalization (e.g., compare CIFAR-10 and CIFAR-100 ticket performance in Figure 3e).

Interestingly, we also observed that when networks are extremely over-parameterized relative to the complexity of the task, as when we apply VGG19 to Fashion-MNIST, we found that transferred winning tickets dramatically outperformed winning tickets generated on Fashion-MNIST itself at low pruning rates (Figure 3b). In this setting, large networks trained on Fashion-MNIST overfit dramatically, leading to very low test accuracy at low pruning rates, which gradually improved as more weights were pruned. Winning tickets generated on other datasets, however, bypassed this problem, reaching high accuracy at the same pruning rates which were untrainable from even Fashion-MNIST winning tickets (Figure 3b), again suggesting that transferred tickets provide additional regularization against overfitting.

Finally, we observed that winning ticket transfer success was roughly similar across across VGG19 and ResNet50 models, but several differences were present. Consistent with previous results [8], performance on large-scale datasets began to rapidly degrade for ResNet50 models when approximately 5-10% of weights remained, in contrast to VGG19 models which only demonstrated small decreases in accuracy, even with 99.9% of weights pruned. In contrast, at extreme pruning fractions on small datasets, ResNet50 models consistently achieved similar or only slightly degraded performance relative to over-parameterized models (e.g., Figures 4c and 4d). These results suggest that ResNet50 models may have a sharper "pruning cliff" than VGG19 models.

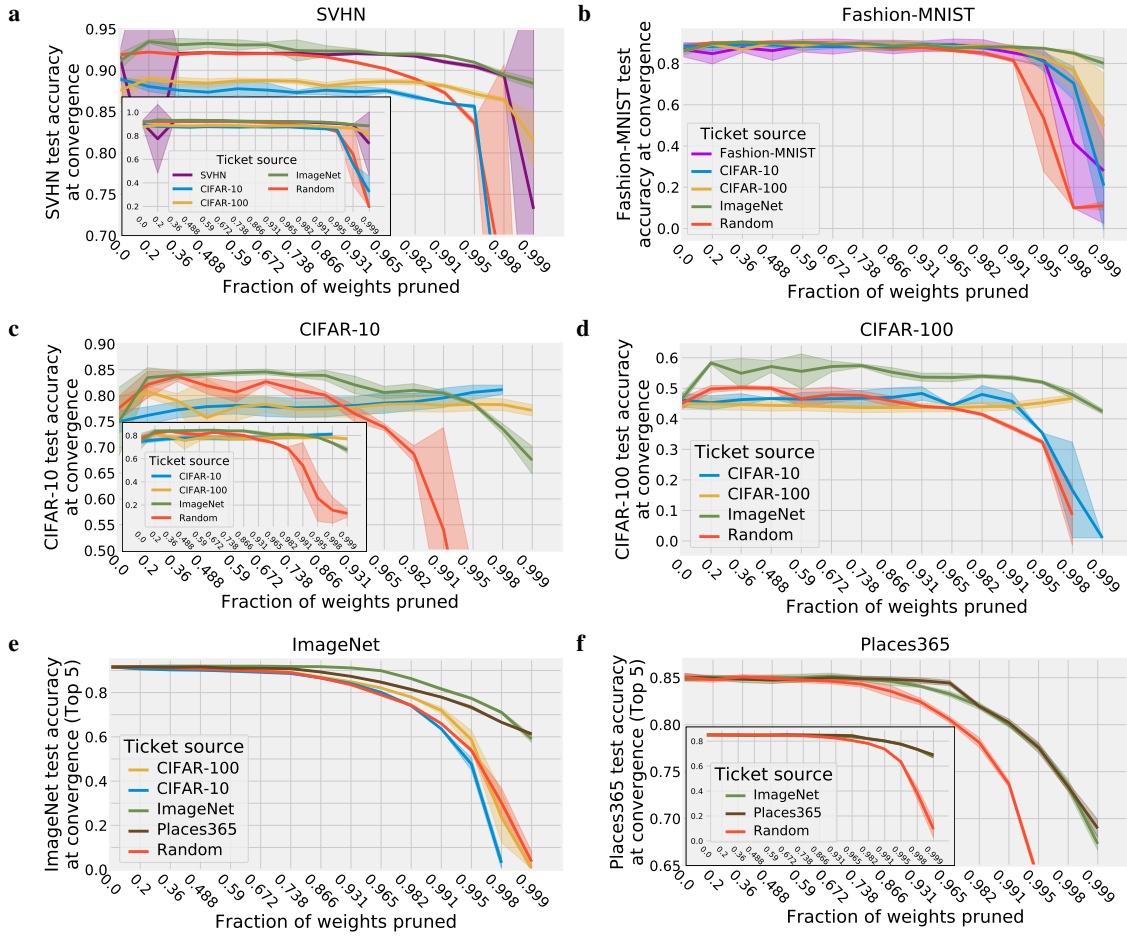

**Figure 4: Transferring ResNet50 winning tickets across datasets.** Winning ticket performance on target datasets: SVHN (**a**), Fashion-MNIST (**b**), CIFAR-10 (**c**), CIFAR-100 (**d**), ImageNet (**e**), and Places365 (**f**). Within each plot, each line represents a different source dataset for the winning ticket. In cases where the y-axis has been narrowed to make small differences visible, full y-axes are provided as insets. Error bars represent mean ± standard deviation across six random seeds.

## 4.3 Transfer across optimizers

In the above sections, we demonstrated that winning tickets can be transferred across datasets within the same domain, suggesting that winning tickets learn generic inductive biases which improve training. However, it is possible that winning tickets are also specific to the particular optimizer that is used. The starting point provided by a winning ticket allows a particular optimum to be reachable, but extensions to standard stochastic gradient descent (SGD) may alter the reachability of certain states, such that a winning ticket generated using one optimizer will not generalize to another optimizer.

To test this, we generated winning tickets using two optimizers (SGD with momentum and Adam [16]), and evaluated whether winning tickets generated using one optimizer increased performance with the other optimizer. We found that, as with transfer across datasets, transferred winning tickets for VGG models achieved similar performance as those generated using the source optimizer (Figure 5). Interestingly, we found that tickets transferred from SGD to Adam under-performed random tickets at low pruning fractions, but significantly outperformed random tickets at high pruning fractions (Figure 5b). Overall though, this result suggests that VGG winning tickets are not overfit to the particular optimizer used during generation, suggesting that VGG winning tickets are *optimizer-independent*.

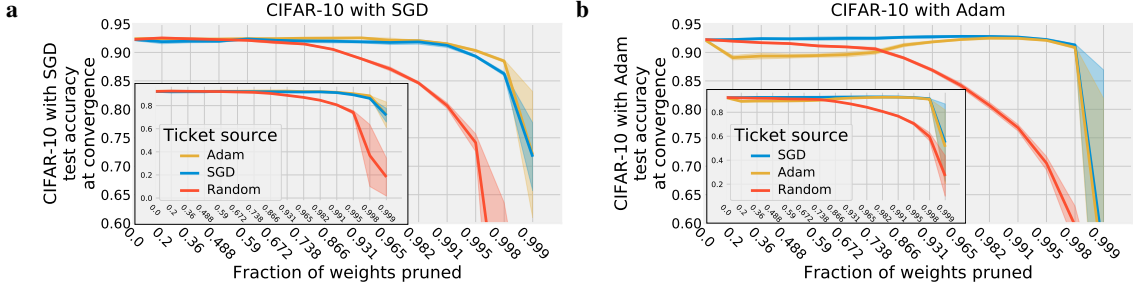

**Figure 5: Transferring VGG winning tickets across optimizers.** Ticket performance when training using SGD w/ momentum (**a**) and Adam (**b**). Within each plot, each line represents a different source optimizer for the winning ticket. In cases where the y-axis has been narrowed to make small differences visible, full y-axes are provided as insets. Error bars represent mean ± standard deviation across six random seeds.

## 5 Discussion

In this work, we demonstrated that winning tickets are capable of transferring across a variety of training configurations, suggesting that winning tickets drawn from sufficiently large datasets are not overfit to a particular optimizer or dataset, but rather feature inductive biases which improve training of sparsified models more generally (Figures 3 and 4). We also found that winning tickets generated against datasets with more samples and more classes consistently transfer better, suggesting that larger datasets encourage more generic winning tickets. Together, these results suggest that winning ticket initializations satisfy a necessary precondition (generality) for the eventual construction of a lottery ticket initialization scheme, and provide greater insights into the factors which make winning ticket initializations unique.

### 5.1 Caveats and next steps

The generality of lottery ticket initializations is encouraging, but a number of key limitations remain. First, while our results suggest that only a handful of winning tickets need to be generated, generating these winning tickets via iterative pruning is very slow, requiring retraining the source model as many as 30 times serially for extreme pruning fractions (e.g., 0.999). This issue is especially prevalent given our observation that larger datasets produce more generic winning tickets, as these models require significant compute for each training run.

Second, we have only evaluated the transfer of winning tickets across datasets within the same domain (natural images) and task (object classification). It is possible that winning ticket initializations confer inductive biases which are only good for a given data type or task structure, and that these initializations may not transfer to other tasks, domains, or multi-modal settings. Future work will be required to assess the generalization of winning tickets across domains and diverse task sets.

Third, while we found that transferred winning tickets often achieved roughly similar performance to those generated on the same dataset, there was often a small, but noticeable gap between the transfer ticket and the same dataset ticket, suggesting that a small fraction of the inductive bias conferred by winning tickets *is* dataset-dependent. This effect was particularly pronounced for winning tickets generated on small datasets. However, it remains unclear which aspects of winning tickets are dataset-dependent and which aspects are dataset-independent. Working to understand this difference, and investigating ways to close this gap will be important future work, and may also aid transfer across domains.

Fourth, we only evaluated situations where the network topology is fixed in both the source and target training configurations. This is limiting since it means a new winning ticket must be generated for each and every architecture topology, though our experiments suggest that a small number of layers may be re-initialized without substantially damaging the winning ticket. As such, the development of methods to parameterize winning tickets for novel architectures will be an important direction for future studies.

Finally, a critical question remains unanswered: what makes winning tickets special? While our results shed a vague light on this by suggesting that whatever makes these winning tickets unique is

somewhat generic, *what precisely* makes them special is still unclear. Understanding these properties will be critical for the future development of better initialization strategies inspired by lottery tickets.

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
