[Supplementary Material]

# A  Appendix

## A.1  Model details and hyperparameters

**General hyperparameters**    All models were trained in PyTorch [26]. For Fashion-MNIST, SVHN, CIFAR-10, and CIFAR-100, batch sizes of 512 parallelized across 2 GPUs were used. For ImageNet and Places365, batch sizes of 512 parallelized across 16 GPUs were used and trained using synchronous distributed training. For models trained with SGD, a learning rate of 0.1 was used with momentum of 0.9 and weight decay of 0.0001. For models trained with Adam, a learning rate of 0.0003 was used with betas of 0.9 and 0.999 and weight decay of 0.0001.

**VGG19**    VGG19 was implemented as in [30], except all fully-connected layers were removed and replaced with an average pool layer, as in [7, 8]. Precisely, filter sizes were as follows: {64, 64, max-pool, 128, 128, max-pool, 256, 256, 256, 256, max-pool, 512, 512, 512, 512, max-pool, 512, 512, 512, 512, global-average-pool}. ReLU non-linearities were applied throughout and batch normalization was applied after each convolutional layer. For all convolutional layers, kernel sizes of 3 with padding of 1 were used, and all convolutional layers were initialized using the Xavier normal initialization with biases initialized to 0, and batch normalization weight and bias parameters initialized to 1 and 0, respectively. All VGG19 models were trained for 160 epochs. Learning rates were annealed by a factor of 10 at 80 and 120 epochs.

**ResNet50**    ResNet50 was implemented as in [15]. Precisely, blocks were structured as follows (stride, filter sizes, output channels): (1x1, 64, 64, 256) x 3, (2x2, 128, 128, 512) x 4, (2x2, 256, 256, 1024) x 6, (2x2, 512, 512, 2048) x 3, followed by an average pool layer and a linear classification layer. All ResNet50 models were trained for 90 epochs. Learning rates were annealed by a factor of 10 at 50, 65, and 80 epochs.

**Pruning parameters**    For all models, an iterative pruning rate of 0.2 was used and 30 pruning iterations were performed. Following the sixth pruning iteration, model performance was evaluated every third pruning iteration. Pruning was performed using magnitude pruning, such that the smallest magnitude weights were removed first, as in [12, 19]. For Fashion-MNIST, SVHN, CIFAR-10 and CIFAR-100 winning tickets, late resetting of 1 epoch was used. For ImageNet and Places365 winning tickets, late resetting of 3 epochs was used.

## A.2  Randomized masks

**Figure A1: Comparison of different random masks.** Performance of various random masks on CIFAR-10. Error bars represent mean $\pm$ standard deviation across six random seeds.

When models are pruned in an unstructured manner, there are two aspects of the final pruned model that may be informative: the values of the weights themselves and the structure of the mask used for pruning. In the original lottery ticket study [7], bad tickets were compared with randomly drawn values, but the preserved winning ticket mask. This has the unintended consequence of transferring information from the winning ticket to the bad ticket, potentially inflating bad ticket performance. This issue seems particularly relevant given that we observed that global pruning results in substantially better performance and noticeably different layerwise pruning ratios relative to layerwise pruning (Figure 1), suggesting that the mask statistics likely contain important information.

To test this, we evaluated three types of masks: preserved masks, globally permuted masks, and locally permuted masks. In the preserved mask case, the same mask found by the winning ticket is used for the random ticket. In the locally permuted case, the mask is permuted within each layer, such that the exact structure of the mask is broken, but the layerwise statistics remain intact. Finally, in the globally permuted case, the mask is permuted across all layers, such that no information should be passed between the winning ticket and the bad ticket.

Consistent with the lottery ticket hypotehsis, we found that the winning ticket outperformed all random tickets (Figure A1). Interestingly, we found that while locally permuting masks damaged performance somewhat (blue vs. green), globally permuting the mask results in dramatically worse performance (blue/green vs. yellow), suggesting that the layerwise statistics derived from training the over-parameterized model are very informative. As this information would not be available without going through the process of generating a winning ticket, we consider the purely random mask to be the most relevant comparison to training an equivalently parameterized model from scratch.