[Reviews · NeurIPS 2019]

Reviewer 1



Update after author reply: The author response was okay, I won't update my score. I agree with the other reviewers: 1. rescaling the weights during initialization should be a part of the experimental setup, that might change these results. that should be a central part of these experiments, and the authors should update the paper with such experiments. 2. as is common in lottery ticket work, there are insufficient comparisons against other (non-lottery ticket) approaches. eventually lottery ticket work will have to compare against other relevant work; for now, i don't think this means we should reject. My original review is below ================================================ The experimental results are convincing, and the experimental setup is not bad. This paper is original, and answers a relevant and timely question for lottery ticket research. It's clearly written, and while it does leave a few questions about exact implementation details the author do a better job than most at being clear. This feels like a complete result, with thorough experiments. One problem throughout research on lottery tickets is the lack of comparisons against other pruning methods. It's understandable that this work only analyzes lottery tickets, as already this includes a lot of experimentation. However, it's not clear to me if the properties found here are simply re-finding the same generalization properties that have been known about other pruning methods (e.g. L1 or L0 feature selection) for years.

Reviewer 2



***POST-REBUTTAL*** Thank you for the time you spent writing the rebuttal! I think that the finding that LT can generalise (I use the word "can" because it does not seem that this is true consistently) is an interesting one, and with some changes, this paper would deserve publication at a top venue like NeurIPS. However, I think we still see things differently on two points. Firstly, I do not believe that comparison to existing algorithms is orthogonal to the topic of this paper. You claim that "... we may be able to generate new initialization schemes which can substantially improve training of neural networks from scratch" and I agree, but the point I am making is that there are other ways of obtaining a better initialisation (e.g., unsupervised pretraining and/or layer-wise pretraining) which are known to improve performance and speed up converge, some of them using less computation than is required to generate a lottery ticket. I view your algorithm as yet another way of generating a good init using some data which yields good performance, potentially with other benefits like compression, after some amount of fine-tuning (the fact that LT is trained from scratch and thus require more fine-tuning than using trained weights seems like a drawback, not advantage, from this viewpoint). As R1, I thus have the feeling that your paper is just rediscovering phenomena which are known except for LTs (e.g., using weights pretrained on Imagenet as init for other datasets is a pretty standard practice by now because it often leads to better results and faster convergence). This could still be interesting if LTs were providing better performance or some other advantage (like compression) compared to other relevant algorithms (unsupervised pretraining, pruning, etc.). Unfortunately, I cannot assess if such advantage exists without comparison to other existing algorithms. Furthermore, since you are using standard datasets (Imagenet, CIFAR-10, etc.), it seems that you could provide comparison to other algorithms (e.g., in terms of accuracy and fraction of pruned weights) without any additional compute. Secondly, I still have some doubts about the results you report for random initialisation. In particular, you say: "... the relevant comparison here is between winning tickets and random tickets neither of which is rescaled", and that "We therefore consider it unlikely that rescaling would change our core results since we have no reason to expect that rescaling would preferentially benefit winning or random tickets." I am not sure about the winning but for the random tickets, the scale does matter (especially when over 90% of the weights are pruned)---please see the He et al., Xavier et al., and other well-cited papers about initialisation of deep neural networks which show that dependent on the scale of the init, the final performance can vary a lot. I suspect that the randomly initialised networks are underperforming (at least partly) because their scale is way off the commonly used 1 / sqrt(no. of inputs) or 1 / sqrt(no. of inputs + no. of outputs), and it may be that this does not have such a strong effect on LTs because they are picked by magnitude based pruning, and thus have naturally higher magnitude than random init. I may be wrong but without additional experiments, I am unable to accept the "core results" in your paper at their face value. ***ORIGINAL REVIEW*** This paper studies whether lottery tickets generalise between datasets and optimisers. Since generation of well-performing lottery tickets, at least for large datasets, is very computationally expensive (the models are retrained up to 20 or 30 times), finding a method that allows us to only go through this process only once and then transfer to other datasets might extend applicability of the lottery tickets for network pruning (albeit it must be said that the presented algorithm does not allow to transfer between different architectures). Two major concerns remained in my mind after reading this paper. Firstly, since the end use of your algorithm is sparsification, I am really missing comparison with alternative algorithms that speed up computation at prediction time (you cite some alternative sparsification approaches in the related work section; other alternatives include the line of work exploiting low precision computation, like “Binarized Neural Networks”, “XNOR-Nets”, etc.). Secondly, even though the original lottery ticket hypothesis is certainly interesting, it seems like “late resetting” turns the algorithm of obtaining the “lottery ticket” into more of a pruning technique, since it essentially makes the initialisation data and “large network” dependent (due to the dependence of the initial value on the first few training iterations of the large network). Hence, viewing your algorithm as a pruning technique, it seems like it should be possible to save some computation and potentially obtain better results by doing transfer learning on top of the **already trained** pruned architecture. Of course, I might be completely wrong, but I would have liked to see the comparison or at least some discussion of this alternative in relation to your algorithm. I am thus skewing towards recommending rejection of this paper at this time. Major comments: - Can you please clarify lines 150-155? In particular, I do not understand if and how do you use the “target” dataset when generating the lottery ticket on the “source” dataset. - Are the weights in any way rescaled after each pruning so that the scale of outputs are approximately preserved? - You explain how differing number of outputs has been handled in the transfer experiments, but I have missed an explanation of how differences in the input dimension are handled?! - How exactly is “global pruning” executed? Specifically, weights in each layer are at initialisation of the scale 1 / sqrt(no. of inputs to the layer) and since there is a growing body of evidence showing that this property is approximately preserved throughout training, magnitude based pruning will preferentially prune weights in the large layers. This is in line with your observations on the bottom of page 3, but I am not entirely sure this is a desirable behaviour. Have you tested what would happen if you pruned based on the rescaled magnitude (e.g., if the initial weight value is w = \sqrt(2 / no. of inputs) \eps, then you could only prune based on the value of \eps instead of w)? Minor comments: - At several places, you cite [7] when [8] should be cited and vice versa (e.g., on p.3, you cite [7] for late resetting). - [push back a little on the generality claims: “... suggesting that [winning tickets] are not overfit to a particular optimizer or dataset” -- clearly, the tickets from smaller datasets fared similarly to random on several datasets + the ResNet example (Fig. 2) and Fig. 3b show that the effects are not universal] - Random masks (p.4): To facilitate comparison, please follow the previous papers and include the results for models with random reinitialisation of the already pruned architecture (i.e. the ‘preserved mask’ curve from the appendix).

Reviewer 3



# Originality To my knowledge, this is the first study of the transfer properties of tickets found in one dataset to a different dataset. # Quality The experiments were clear and provided error bars across six random seeds for the core experiments. # Clarity I found the writing to be clear and it was generally easy to follow the methodology and results. One complaint I have is that the colors in Figure 4 are inconsistent between the subplots. This made it more difficult for me to follow the patterns that were being pointed out. # Significance I think that the most significant aspect of this is the study of the transfer properties of tickets found in one dataset to a different dataset. Transfer learning is increasingly important as the cost of training large models grows, and finding ways to sparsify the model could help open the way for faster research.

[Author Response · NeurIPS 2019]

We thank the reviewers for their time and valuable feedback on our paper. We've responded to comments below:

**Motivation and Contribution:** In this study, our primary goal was to understand whether winning tickets contain generic inductive biases which apply to multiple problems or rather, whether winning tickets are simply overfit to the particular dataset and optimizer used to generate them. One of the most exciting aspects of the lottery ticket hypothesis is that it suggests that we may be able to generate new initialization schemes which can substantially improve training of neural networks from scratch without requiring any pruning or fine-tuning. However, such initialization schemes are only possible if winning tickets contain generic inductive biases. We therefore view the primary contribution of this study as a scientific one – understanding the generality of winning tickets – rather than a prescriptive one, though we hope that this study will help to lay the groundwork for improved training methods in the near future. Importantly, the absolute performance of winning tickets is independent of the generality of winning tickets.

**Comparisons against other pruning methods (R1, R2):** We completely agree with the reviewers that comparisons between winning ticket trained networks and more traditional model compression approaches are interesting and important problems, but we feel this an orthogonal question to the primary question of this study (how generic are winning tickets), especially since these comparisons are difficult to make fairly. Pruning approaches generally require the full model to be trained from scratch, after which the model is generally iteratively pruned and fine-tuned. In contrast, winning tickets (especially transferred winning tickets) can be trained from scratch, and do not require any further fine-tuning. As such, the costs of generating these models are quite different, making fair comparisons challenging. We will add a discussion point clarifying these differences in section 5.1.

As Reviewer 1 acknowledged, our study required massive amounts of compute, especially since we performed six replicates of each experiment. For example, we trained over 1500 ImageNet/Places365 models from scratch to generate the results in Figs. 3e,f and 4e,f. We therefore view such comparisons as beyond the scope of the present work.

**Analysis of lottery tickets (R1)** : We wholeheartedly agree with the reviewer that this an exciting and important line of inquiry (and one that we are currently pursuing!). However, detailed analysis of the structure of winning tickets is quite challenging since winning tickets and random tickets are often statistically quite similar (at least to the first few moments). We also agree that comparing pruning patterns to those derived from $L_0$ or $L_1$ regularization are interesting, but we haven't analyzed these yet. As such, we leave this problem for future work.

**Rescaling of weights (R2):** To be clear, after each pruning iteration, a new mask is generated, after which the subsequent model is trained from scratch with the new mask. Because converged weights are not used after pruning (as in typical compression approaches), there is no need to re-scale the weights. Additionally, the relevant comparisons for our primary question are between the transfer ticket performance, same dataset ticket performance, and random ticket performance, all of which have the same scaling of initialization.

**Differences in input dimension (R2):** For both the VGG19 model and the ResNet50, a global average pool is applied across all spatial dimensions prior to the final linear output layer. As a result, changes in input dimension do not require changes in model architecture. We will add a paragraph clarifying this point within section 3.3 in the final paper.

**Global pruning scaling (R2):** We have not evaluated rescaling the weights based on the pruning fraction. However, the relevant comparison here is between winning tickets and random tickets neither of which is rescaled. We therefore consider it unlikely that rescaling would change our core results since we have no reason to expect that rescaling would preferentially benefit winning or random tickets.

**Clarification of target/source in lines 150-155 (R2):** In order to evaluate the generality of winning tickets, winning tickets were generated by iteratively training and pruning a model on one dataset/optimizer ("source" configuration, as defined in lines 144-145), and evaluated on a second dataset/optimizer ("target" configuration). For standard lottery ticket experiments in which the source and target dataset are identical, each iteration of training represents the winning ticket performance for the model at the current pruning fraction. However, because the source and target dataset/optimizer are different for our experiments and because we primarily care about performance on the target dataset for this study, we must re-evaluate each winning ticket's performance on the target dataset, adding an additional training run for each pruning iteration. In the final paper, we will expand this section to better clarify this point.

**Inconsistent colors (R1, R3):** Thank you for pointing out the inconsistency in colors between plots! We agree that this makes the plots difficult to read and will correct the colors to be consistent across plots and figures in the final paper.

**Inclusion of preserved mask (R2):** While we feel the use of globally random masks is important to properly evaluate the performance of random tickets (because masks can leak substantial amounts of information from the final trained model to the initialization, as discussed in Section 3.1), we will add additional plots with the preserved mask case to the appendix for our primary results in the final paper.

**Generality claims (R2):** We agree with the reviewer that this claim is a little too strong since, indeed, some winning tickets generated on small datasets do not generalize to larger datasets. We will soften this claim in the final paper.

[Meta-Review · NeurIPS 2019]

This paper is close to a borderline but I think it should be accepted. The question in the paper, whether the lottery tickets are transferable across datasets, is a very obvious one to ask and I am certain that many people are thinking about it. I think there is a value in getting this paper out so that others can build upon these results. The work is mostly well executed and, with minor exceptions, it is clearly written. While suggesting this paper’s acceptance, I do suggest that the authors run the experiments suggested by the reviewer 2 and put the results in the camera-ready version.